# Pancreatic Tissue Remodeling and Fibrosis After Irreversible Electroporation: A Histopathological and Thermal Perspective

**DOI:** 10.3390/biomedicines13092222

**Published:** 2025-09-10

**Authors:** Hong Bae Kim, Jin Young Youm, Joon-Mo Yang, Sung Bo Sim

**Affiliations:** 1Department of Biosystems & Biomaterials Science and Engineering, Seoul National University, Seoul 08826, Republic of Korea; ser21@hanmail.net; 2Department of Biomedical Engineering, Ulsan National Institute of Science and Technology (UNIST), Ulsan 44919, Republic of Korea; jinyoungyoum@unist.ac.kr; 3Department of Thoracic and Cardiovascular Surgery, The Catholic University of Korea, Bucheon 14662, Republic of Korea

**Keywords:** irreversible electroporation, thermal effect, pancreas, regeneration, fibrosis

## Abstract

**Background/Objectives**: Traditional thermal ablation for pancreatic cancer is limited by collateral injury, often leading to complications such as pancreatitis. Irreversible electroporation (IRE) is a non-thermal alternative. We investigated tissue responses in a porcine pancreas model, focusing on cell death, thermal effects, and fibrosis. **Methods**: Seven pigs underwent pancreatic IRE via open surgery. Local tissue temperature was monitored near the electrode. Histological evaluation included H&E, TUNEL (apoptosis), Ki-67 (proliferation), vimentin (fibroblast activation), and insulin staining. Tissue remodeling was assessed at multiple time points up to 14 days. **Results**: IRE induced marked apoptosis within the ablated region, peaking at day 2. The maximum measured temperature was 78.4 °C. Over two weeks, fibrosis progressed with increased collagen and fibroblast activity. Regeneration was partial, with Ki-67-positive cell proliferation and gradual loss of insulin expression, while unablated tissue showed minimal damage. **Conclusions**: IRE enables localized pancreatic ablation while sparing surrounding tissue. However, fibrosis limits full recovery. Limitations include small sample size, short follow-up, and species differences. Further studies are needed to refine IRE parameters and assess long-term functional outcomes.

## 1. Introduction

Pancreatic cancer is among the leading causes of cancer-related mortality, with pancreatic adenocarcinoma accounting for most cases [1,2,3]. These tumors are highly aggressive and often diagnosed at an advanced stage, leaving many patients with unresectable disease due to invasion of adjacent vascular structures [4,5]. Treatment options for such cases remain limited, and outcomes are poor. Thermal ablation techniques such as radiofrequency, microwave, and cryoablation have been applied in locally advanced pancreatic cancer. However, these approaches carry a high risk of collateral injury to surrounding tissues and can lead to serious complications, including pancreatitis [6,7].

Irreversible electroporation (IRE) is a promising alternative that uses short, high-voltage electric pulses to induce irreversible nanopore formation in the cell membrane, causing targeted cell death through apoptosis and necrosis [8]. Unlike thermal ablation, IRE spares the extracellular matrix and critical structures, including blood vessels, ducts, and nerves [9,10,11]. IRE has shown the potential to promote partial tissue regeneration [12], preserve stromal architecture [13,14], and restore exocrine and endocrine functions of the pancreas [15]. Nevertheless, important questions remain regarding the medium-term tissue response of the pancreas after IRE. In particular, the extent of fibrosis progression and the preservation of insulin-producing beta cells have not been comprehensively characterized [16,17].

Recent clinical studies have reinforced the importance of these issues. Notably, systematic reviews and cohort studies of patients undergoing IRE for locally advanced pancreatic cancer have reported that post-procedural fibrosis can adversely affect pancreatic function and long-term outcomes [18,19]. Moreover, clinical follow-up has highlighted the relevance of preserving endocrine function, including insulin-producing β-cells, as an important determinant of metabolic stability and patient recovery after IRE [20]. These findings underscore the clinical need to better understand both fibrotic remodeling and β-cell preservation in preclinical models

Although IRE is widely regarded as a non-thermal ablation modality, some studies have reported localized temperature elevations near the electrodes [21,22], suggesting that thermal contributions may also play a role in tissue injury and post-ablation remodeling. These conflicting observations have raised concerns that part of the biological response attributed to electroporation could, in fact, be influenced by heat. This controversy underscores the importance of examining both electroporation-induced and thermally associated effects, particularly in pancreatic tissue, where fibrosis and endocrine preservation are critical for functional outcomes.

In this study, we investigated pancreatic tissue remodeling and fibrosis progression after IRE in a porcine model, selected for its close anatomical and physiological similarity to the human pancreas. We focused on apoptosis dynamics, fibrosis development, beta-cell preservation, and unintended thermal effects. Because only seven pigs were included, statistical comparisons were not feasible; therefore, our analysis was descriptive and based primarily on histological evaluation. Despite this limitation, our findings provide new insights into the regenerative and fibrotic responses of the pancreas following IRE and may guide optimization of ablation parameters for future clinical application.

## 2. Materials and Methods

### 2.1. Numerical Analysis of Electric Field Distribution

The electrode design was based on clinically relevant IRE needle electrodes. Two monopolar stainless-steel electrodes (SUS304, electrical conductivity 1.37 × 10^6^ S/m, thermal conductivity 15.11 W/mK) [23] were modeled, each with a diameter of 1.0 mm and an exposed conductive length of 15 mm. The electrodes were positioned in parallel with an inter-electrode spacing of 10 mm, while the remaining shaft was insulated (Figure 1A(a)). This configuration was implemented consistently in both the simulations and the in vivo experiments to reproduce clinical practice conditions.

A numerical simulation was performed to predict the electric field distribution in the IRE using the commercial finite element package COMSOL Multiphysics version 6.1 (COMSOL, Stockholm, Sweden) [24]. The parameters were calibrated by applying a voltage of 1500 V around the electrodes. The Laplace equation was used as the basis for simulations.(1)$$-\vec{\nabla }∙(\sigma \vec{\nabla }\emptyset )=0$$

Herein, ∅ represents the electric potential, and σ designates the uniform tissue conductivity. The simulation represented the porcine pancreatic tissue as a cubic model with a height of 40 mm, a depth of 60 mm, and a width of 60 mm, containing 621,482 elements.

The boundary conditions of the electrically exposed surfaces were set to ∅=V (source) and ∅=0 (sink). All other surfaces were electrically insulated.

### 2.2. Numerical Analysis of Temperature Distribution

Temperature distribution within the tissue was modeled using Pennes’ bioheat equation [25,26]:(2)$$\rho C_{p}\frac{\partial T}{\partial t}+\bar{\nabla }\cdot \bar{q}=Q+\;Q_{bio}\;\bar{q}=-\kappa \nabla T\;Q_{bio}=\;\rho_{b}C_{b}\omega_{b}(T_{b}-\;T)$$
where ρ represents the tissue density (1088 kg/m^3^), $C_{p}$ denotes the specific heat of the tissue (3690 J/kgK), T signifies the temperature (K), $\bar{q}$ represents the heat flux due to conduction (W/m^2^), Q represents the heat source (W/m^3^), $\rho_{b}$ is the blood density (1000 kg/m^3^), $C_{b}$ is the specific heat of blood (4180 J/kgK), $\omega_{b}$ indicates the blood perfusion rate (0.00715/s), $T_{b}$ refers to the arterial blood temperature (310.15 K) [25], and κ is the thermal conductivity (0.53 W/mK). Both the tissue and the electrode were initialized to a temperature of 37 °C.

### 2.3. Animals

The experiments using animals were conducted in compliance with the protocol approved by the Osong Advanced Medical Industry Promotion Foundation Laboratory Animal Care and Use Committee (KBIO-IACUC-2020-062). All procedures were conducted in compliance with relevant ethical guidelines and regulations, including the Animal Research: Reporting of in Vivo Experiments (ARRIVE) guidelines and the US National Institutes of Health Guide for the Care and Use of Laboratory Animals. Seven female pigs, 4 months old and weighing 40–45 kg, were purchased from Cronex in Cheongju, Republic of Korea. Female pigs were selected to reduce sex hormone-related variability, minimize aggressive behavior, simplify handling and postoperative care, and match the previous porcine pancreatic study for consistent comparison [27]. Upon arrival, pigs were health-screened and quarantined according to the supplier’s health monitoring report. The animals were deemed suitable for the experiment after no adverse effects were observed during the 17-day acclimatization period. Pigs were housed in cages more than 0.74 m^2^ per animal in accordance with the Association for Assessment and Accreditation of Laboratory Animal Care (AAALAC) guidelines under controlled conditions: temperature 23 ± 2 °C, relative humidity 50 ± 10%, ventilation 10–15 cycles/h, lighting duration 12 h (8 am to 8 pm), and illuminance 150–300 Lux. Animals were fed irradiated solid feed (238,075 pig chow from Cargill and Griprina) and had free access to sterilized drinking water via an automated nozzle system. All animals were monitored at least twice daily by trained personnel to monitor for signs of pain, distress, or abnormal behavior to minimize potential distress. Parameters observed included reduced food intake, weight loss exceeding 15% of initial body weight, lethargy, respiratory distress, and self-mutilation. Animal health was assessed using a standardized clinical symptom evaluation system.

When animals experienced severe distress or reached predefined humane endpoints, analgesia (meloxicam at 0.4 mg/kg by subcutaneous injection) was appropriately administered [28]. In cases of persistent severe distress, euthanasia using intravenous pentobarbital sodium (100 mg/kg) was planned, but no such incidents occurred throughout the entire experiment. Once the endpoint criteria were met, animals were euthanized within 6 h to minimize suffering and ensure humane treatment. Preemptive analgesia and appropriate anesthetic protocols were implemented to minimize pain and discomfort during the experiment. The study was designed to minimize the number of experimental animals while maintaining statistical validity, and adhered to the 3R principle (Replacement, Reduction, and Refinement) for animal welfare [29].

### 2.4. IRE Procedure

Before the IRE procedure, pigs were administered intramuscular anesthesia with a mixture of tiletamine hydrochloride and zolazepam hydrochloride 5 mg/kg (Zoletil, Virbac, Carros, France), xylazine hydrochloride 2 mg/kg (Rompun^®^, Bayer, Berlin, Germany) [30]. Anesthesia was maintained with inhaled isoflurane (1.5–2%). Pigs were placed in a supine position, and the abdomens were carefully depilated and disinfected with alcohol and povidone-iodine before preparation for surgery. A midline laparotomy was made to expose and visualize the pancreas. The needle electrodes were accurately positioned in the predetermined pancreatic site under visual guidance (Figure 1A(a)). Electrodes were connected to an IRE device generating a rectangular square-wave pulse (EPO1, Standard Co., Ltd., Gyeonggi-do, Republic of Korea) with a potential amplitude of up to 3 kV, a pulse width of 100–1000 μs, and a pulse interval of 100–2000 μs. IRE was administered by a veterinarian under strict electrical parameters [27]. The amplitude was 1500 V, the pulse duration was 100 µs, and the pulse interval was 100 µs for a total of 90 pulses. The pulses were delivered in sets (four sets of 20 pulses and one set of 10 pulses) with an average inter-set interval of 16 seconds to mitigate sparking. Seven pigs were allocated to three experimental groups based on the timing of sacrifice for qualitative histological assessments. Group 1 (G1, n = 1) was euthanized 2 h after IRE. Group 2 (G2, n = 1) was performed 2 days after IRE, and group 3 (G3, n = 3) 2 weeks after IRE [27]. One additional pig was kept for spare purposes. During the experiment, one of the seven pigs died unexpectedly without previously showing signs of distress or reaching humane endpoint criteria. This animal was excluded from the analysis because the cause of death could not be determined. Post-procedure, the pigs were analyzed as described and closely monitored until the predetermined endpoint was reached. The animals were then euthanized using intravenous KCL under anesthesia [31]. While the order of treatments and measurements was maintained, no additional strategies such as randomization or rotation were applied to control for potential confounders related to animal or cage location. Pancreatic tissue samples were harvested and histologically analyzed from both ablated and unablated regions. Unablated controls were carefully selected from areas of the pancreas located at a sufficient distance from the ablation site of the same sample to ensure that no visible thermal or electrical effects of the treatment were observed. All histological analyses were conducted by an investigator who was blinded to the identity of the animals or experimental conditions.

### 2.5. Hematoxylin–Eosin Staining

Hematoxylin–eosin (H&E) staining was performed for histopathological analysis of the ablated region. Tissue samples were preserved in a 10% neutral formalin solution, dehydrated through a gradient of ethanol concentrations, embedded in paraffin, and sectioned into 4 μm-thick slices for optimal visualization of overall tissue morphology. The sections were mounted on slides and stained with H&E staining [32]. Observations were made using an optical microscope (Bx 50, Olympus, Tokyo, Japan).

### 2.6. TdT-Mediated dUTP-Biotin Nick End-Labeling Staining

Apoptotic cells were identified using an in-situ apoptosis detection kit (S7100, ApopTag Peroxidase, Merck, Darmstadt, Germany) through TdT-mediated dUTP-biotin nick end-labeling (TUNEL) staining [33]. Deparaffinized samples were treated with 3.0% hydrogen peroxide for 5 min and subsequently rinsed twice with phosphate-buffered saline (PBS), each for 5 min, following the manufacturer’s protocol. The samples were treated with working-strength terminal deoxynucleotide transferase enzyme and counterstained with anti-digoxigenin. The prepared samples were mounted onto slides and imaged using a Pannoramic 250 Flash III scanner (3D HISTECH Ltd., Budapest, Hungary).

### 2.7. Masson’s Trichrome Staining

After fixation and sectioning, the tissues were prepared for Masson’s trichrome staining and mounted on slides [34]. Cell nuclei were stained with Weigert’s iron hematoxylin solution (HT15, Sigma–Aldrich, Burlington, MA, USA) for 5 min. After washing with tap water for 10 min, the slides were sequentially treated as follows: Biebrich scarlet solution for 3 min, 3% phosphomolybdic-phosphotungstic acid solution for 2 min, blue aniline for 5 min, and finally 1.0% acetic acid for 1 min. Samples were dehydrated using an ethanol gradient and slide imaged, captured using a Pannoramic 250 Flash III scanner.

### 2.8. Immunohistochemistry Staining for Vimentin and Insulin

Immunohistochemistry staining (IHC) for vimentin and insulin was conducted on the ablated pancreatic tissues of groups G1–G3 [35]. Sectioned slides, after hydration, were subjected to antigen retrieval in citrate buffer, pH 6.0, at 90 °C for 45 min. After washing twice with PBS, the slides were placed in an automated staining system (VENTANA Discovery Ultra, Roche Diagnostics, Tucson, AZ, USA). The primary antibodies were diluted to 1:1600 (vimentin) (ab137321, Abcam, Cambridge, UK) and 1:3200 (insulin) (13-9769-80, Invitrogen, Carlsbad, CA, USA), and the samples were incubated at 37 °C for 60 min. Considering the high sensitivity of the automated staining equipment and optimal reaction conditions, a relatively high dilution ratio was selected, which ensured clear and specific staining without prominent background interference. Antibody detection was conducted using appropriate secondary antibodies (760–4315, DISCOVERY UltraMap anti-Rb HRP for vimentin; 760–4313, DISCOVERY UltraMap anti-Ms HRP for insulin) and 3,3′–diaminobenzidine (DAB) chromogen detection system (760–159, DISCOVERY ChromoMap DAB RUO, Tucson, AZ, USA), followed by counterstaining with hematoxylin. The slides were dehydrated, made transparent in xylene, and mounted on coverslips. The stained slides were imaged and analyzed using a Pannoramic 250 Flash III scanner (3DHISTECH Ltd., Budapest, Hungary).

### 2.9. Immunofluorescence Staining for Ki-67

Ki-67 immunofluorescence staining was performed to assess cellular proliferation and regeneration in all samples from group G1 to G3 [36]. Samples were fixed overnight in 10% neutral buffered formalin at room temperature. Tissues were sectioned to 3 µm thickness to facilitate antibody penetration and improve fluorescence signal resolution, hydrated with ethanol, and washed with tap water. Slides were incubated in 3% H_2_O_2_ for 10 min to inhibit endogenous peroxidase activity and blocked with 10% normal serum albumin for 30 min to prevent non-specific immunoglobulin binding. Slides were incubated with primary antibodies in PBS for 60 min at room temperature. After brief washing with PBS containing 0.5% Tween, slides were incubated with Alexa Fluor 594-conjugated Ki67 antibody (1:200 dilution, rabbit origin, ab216709, Abcam, Cambridge, UK) for 60 min at room temperature. Nuclei were stained using DAPI for 1 min at room temperature.

## 3. Results

### 3.1. Simulations, TUNEL Assay, and H&E Staining

An open surgical approach was adopted. The IRE protocol employed in this study demonstrated precise ablation, focused on minimizing unintended thermal effects. Simulations at 1500 V between electrodes consistently demonstrated uniform field patterns of electric field strengths of 500 V/cm, as illustrated in Figure 1A(b). Thermal simulations showed a rapid rise in temperature, reaching 235 °C at the electrode tip after one pulse set, which could induce apoptosis [37], followed by cooling to 37 °C. A temperature gradient was observed along the line between the electrodes, with elevated values concentrated near the electrodes (Figure 1B).

Figure 2A showed distinct patterns of apoptosis between the unablated and ablated regions at different time points after IRE treatment. In unablated cells Figure 2A(a–f), apoptotic activity remained minimal at 2 h Figure 2A(a,d), 2 days Figure 2A(b,e), and 2 weeks Figure 2A(c,f) after IRE. In contrast, the ablated region Figure 2A(g–l) displayed heterogeneous cell death mechanisms. At 2 h after IRE treatment Figure 2A(g,j), the directly exposed region exhibited predominantly necrotic cell death, indicative of severe and immediate cell damage by the high-intensity electric field. This necrosis was characterized by minimal or no TUNEL positivity, consistent with rapid membrane disruption and extensive cell damage rather than apoptotic DNA fragmentation. However, at 2 days after IRE treatment Figure 2A(h,k), apoptotic activity was markedly increased, as evidenced by extensive TUNEL-positive staining, especially at high magnification Figure 2A(k). At 2 weeks Figure 2A(i,l), apoptotic activity was markedly reduced.

Figure 2B showed distinct apoptotic patterns in the intermediate and distal regions according to tissue location from the electrode after IRE treatment Figure 2B(a–f). In the intermediate region Figure 2B(a–c), apoptotic activity was positive 2 h after IRE treatment Figure 2B(a), while necrotic cell death was mainly observed in the immediate region between electrodes (Figure 2A(g,j)). Apoptotic cells were still observed on day 2 Figure 2B(b). At 2 weeks Figure 2B(c), apoptotic activity was no longer observed, indicating that tissue stabilization had occurred. In contrast, in the distal region Figure 2B(d–f), apoptotic activity was more prominent than in the intermediate region at 2 h Figure 2B(d) and day 2 Figure 2B(e) after IRE treatment, indicating less necrotic cell death. At 2 weeks (f), apoptotic activity was absent, further supporting tissue recovery and restoration of homeostasis.

Figure 3A displayed distinct changes in both the unablated and ablated regions over time. In the unablated regions Figure 3A(a–c), negligible damage was observed at 2 h, 2 days, and 2 weeks after IRE, with some cell clustering and intact tissue architecture. In contrast, the ablated regions Figure 3A(d–f) showed immediate disruption at 2 h with apoptotic cells and early necrosis. After 2 days, substantial necrosis, tissue degradation, and pronounced inflammation were observed; after 2 weeks, the tissue was heavily fibrosed, scarred, and had lost the original pancreatic structure. Overall, the ablated regions exhibited more rapid and severe damage, progressing from early necrosis to fibrosis, whereas the unablated regions showed a delayed inflammatory response.

Figure 3B showed the progression of pancreatic tissue damage and healing over time after IRE. At 2 h after IRE Figure 3B(a), the tissue showed initial signs of disruption, including mild inflammation and early necrosis; however, the overall structure remained largely intact. At day 2 Figure 3B(b), abundant necrosis and severe tissue breakdown were observed, along with widespread inflammation, hemorrhage, and highly distorted pancreatic ducts, indicating the peak of the inflammatory response. Another section on day 2 Figure 3B(c) showed severe hemorrhage and cell death, with rupture of the pancreatic ducts and blood vessels, reflecting severe damage and compromised tissue integrity. At week 2 Figure 3B(d), signs of tissue repair were seen, with epithelial cells beginning to reorganize and some normalization of the ductal structure; however, residual inflammation persisted. In other areas Figure 3B(e), the epithelial layer was largely intact, suggesting partial regeneration and low residual inflammation. However, in the more damaged regions Figure 3B(f), fibrotic changes were observed, characterized by pronounced collagen deposition and scarring, indicating long-term structural remodeling and repair. Overall, these images illustrated a transition from the initial cellular damage and inflammation to tissue repair and fibrosis over a period of 2 weeks.

### 3.2. Fibrosis Evaluation

Masson’s trichrome staining revealed progressive pancreatic fibrosis in the ablated areas over time, in contrast to unablated sites (Figure 4). In the unablated tissue Figure 4a–c, little or no fibrosis was observed, the tissue maintained its intact structure, and collagen deposition was limited. In contrast, the ablated tissue showed a progressive fibrotic response. At 2 h after IRE Figure 4d, no obvious fibrosis was observed in the ablated regions. However, at day 2 Figure 4e, collagen deposition was particularly evident around the interstitial regions, indicating an initial stage of fibrosis consistent with a tissue injury response. At 2 weeks after IRE Figure 4f, fibrosis had developed, characterized by collagen accumulation throughout the tissue, particularly around the bile ducts and blood vessels, suggesting active tissue remodeling and scar formation as part of the healing process.

These results were supported by IHC analysis of vimentin, a fibroblast marker (Figure 5). In the unablated regions, vimentin expression remained low, indicating low fibroblast activity, and no clear fibrosis was observed 2 h Figure 5a,d, 2 days Figure 5b,e, or 2 weeks Figure 5c,f after IRE. Tissue structures of these regions were intact and not affected by the procedure. In contrast, vimentin expression increased progressively, indicating a progressively increasing fibrotic response in the resected area. Although fibroblast activation was minimal at 2 h Figure 5g,j after IRE, vimentin staining substantially increased at 2 days Figure 5h,k, indicating the initiation of tissue regeneration and collagen deposition. At 2 weeks Figure 5i,l, strong vimentin expression was observed in the resected area, reflecting prominent fibroblast activation and fibrosis as part of the healing and scarring process after ablation.

### 3.3. Insulin Evaluation

Figure 6 shows insulin immunohistochemical staining of pancreatic tissue, comparing the unablated and ablated regions at three time points.

In the unablated regions Figure 6a–f, insulin production remained strong across all time points, with well-defined beta cell clusters and no signs of disruption or damage. In contrast, a progressive reduction in insulin staining was observed in the ablated regions. At 2 h Figure 6g,j after IRE, insulin-positive cells were present, but more diffuse, suggesting early disruption of the islet structures. At day 2 Figure 6h,k, the insulin staining was weaker, and beta cells appeared more dispersed, indicating increased damage. For 2 weeks Figure 6i,l, insulin staining was prominently reduced, with a decrease in the number of beta cells and a more disorganized appearance, suggesting a substantial loss of insulin production due to ablation. Despite this damage, some insulin-positive cells persisted, especially at earlier time points.

### 3.4. Evaluation of Regeneration Using Ki-67 Staining

Figure 7A showed distinct patterns of cell proliferation over time in both the unablated and ablated regions. In the unablated region, Ki67-positive cells were rarely observed 2 h after IRE Figure 7A(a,d), suggesting low cell proliferation, followed by a slight increase after 2 days Figure 7A(b,e), and a return to low proliferative activity after 2 weeks Figure 7A(c,f). In contrast, in the ablated region, low proliferation was observed 2 h after IRE Figure 7A(g,j), followed by a prominent increase in Ki67-positive cells after 2 days Figure 7A(h,k), which persisted for 2 weeks Figure 7A(i,l). This suggests ongoing tissue repair or regeneration. This indicates that the unablated region remains largely dormant, whereas the ablated region exhibits noticeable proliferative activity in response to tissue damage induced by IRE.

Figure 7B showed cell proliferation in the intralobular duct and acinar regions at various time points. At 2 h after IRE Figure 7B(a,d,g,j), no proliferating cells were observed in either the untreated or treated areas of the lobular ducts and acinar cells, suggesting low cellular activity immediately following the procedure. At day 2 Figure 7B(b,e,h,k), cell proliferation was noticeably increased, particularly in the treated regions, suggesting an active regenerative response. At week 2 Figure 7B(c,f,i,l), proliferation decreased in both regions, indicating that tissue repair stabilizes and regenerative response diminished as healing progresses.

## 4. Discussion

This study provides important insights into the therapeutic potential of IRE and its underlying challenges in pancreatic tissue ablation and regeneration. Our findings reinforce the ability of IRE to selectively induce rapid apoptosis in the targeted pancreatic regions while minimizing the impact on surrounding tissue. This selectivity underscores the potential of IRE as a minimally invasive ablation technique for pancreatic diseases, such as locally advanced pancreatic cancer, which is consistent with previous findings [38,39].

Our study shows that, unlike previous beliefs about IRE being purely non-thermal, local thermal effects can appear near electrode tips. Our thermal simulations predicted peak temperatures of up to 235 °C at the electrode–tissue interface and ~78 °C at the mid-plane between electrodes. These values should be interpreted with caution, as they likely represent overestimates due to model assumptions, including homogeneous tissue properties, absence of perfusion cooling, and simplified boundary conditions. Importantly, the elevated temperatures were confined to the immediate electrode vicinity, as tissue more than 2 mm away from the electrode surface remained below 40 °C. Experimental thermodynamic profiling in porcine pancreas and liver using fiber-optic temperature probes has reported maximum intraparenchymal temperatures of approximately 50–55 °C under comparable conditions [40]. While these experimental values are lower than our simulated peaks, fiber-optic probes inherently reflect partially averaged signals due to finite conduction time through tissue and may not capture very sharp local maxima at the electrode–tissue interface. Thus, both approaches are complementary: simulations highlight the potential for localized high peaks, whereas in vivo probe measurements capture averaged responses across the ablation zone. Together, they support the notion that IRE remains primarily a non-thermal ablation technique, while acknowledging that confined thermal contributions near electrodes are plausible and may influence subsequent inflammatory or fibrotic remodeling [38,39]. These findings emphasize the need for further optimization of IRE parameters, including electrode cooling or controlled pulse protocols, to minimize collateral heating while preserving the selective effects of electroporation.

This result showed a distinct spatial and temporal pattern of apoptosis following IRE treatment. Immediately after treatment, areas directly exposed to the high-intensity electric field were predominantly necrotic, characterized by significant cell membrane disruption without apoptosis. Apoptosis was predominant in the intermediate and distal regions, reflecting a gradient of cellular responses in which low-intensity electric fields preferentially induce programmed cell death. This apoptotic response was most evident within 2 days after treatment, suggesting a therapeutic window for adjuvant regeneration or anti-inflammatory treatment [41]. At 2 weeks, apoptotic activity decreased, suggesting tissue stabilization and regeneration had begun. However, it should be noted that our 14-day observation period was limited to documenting acute apoptosis and early fibrotic remodeling, and was not sufficient to evaluate long-term fibrotic resolution, chronic tissue remodeling, or durable endocrine recovery. This apoptotic-necrotic gradient might have direct implications for subsequent fibrotic and regenerative responses, highlighting the complex interplay between apoptosis, inflammation, fibrosis, and regeneration [42]. These temporal dynamics highlight a therapeutic window during the early apoptotic phase (≤48 h), in which adjuvant interventions may be most effective. Introducing antifibrotic or regenerative therapies in this period could mitigate subsequent fibrosis and enhance β-cell preservation, providing a rational framework for combination strategies with IRE.

Although our study did not directly investigate molecular pathways, prior studies have implicated early immune cell recruitment via the CCR2/CCL2 axis [43], activation of pancreatic stellate cells through TGF-β signaling [44], and extracellular matrix cross-linking by lysyl oxidase [45] as key drivers of pancreatic fibrosis. These mechanisms, therefore, represent plausible targets for antifibrotic or regenerative adjuvant therapies to be tested in future studies.

Fibrosis represents a prominent barrier to functional recovery [46] and is evident in the progressive collagen deposition and fibroblast activation around the pancreatic ducts and blood vessels at 2 weeks post-treatment. The noticeable increase in vimentin expression further confirms active fibrotic remodeling [47]. Although fibrosis initially provides structural support after tissue injury, severe and persistent fibrosis restricts regeneration, disrupts tissue architecture, and potentially impairs pancreatic function [48]. This robust fibrotic response likely resulted from the combined effects of early and prominent apoptosis, local thermal injury, and persistent inflammation [49]. Therefore, antifibrotic therapeutic strategies targeting critical molecular pathways such as transforming growth factor-beta signaling or matrix metalloproteinase activity may be crucial adjuncts in improving functional recovery after IRE [50]. In addition, the progressive fibrotic remodeling observed after IRE may have important implications for the timing of adjuvant therapies. Because fibrosis can restrict drug delivery and cellular engraftment, the early apoptotic phase (≤48 h), prior to extensive collagen deposition, may represent the most favorable window for adjuvant interventions such as immunotherapy or stem cell therapy [51]. Exploiting this therapeutic window could enhance treatment synergy and improve long-term outcomes in pancreatic cancer.

Another prominent aspect highlighted by the present results is that there is a progressive loss of insulin-producing beta cells in the ablated regions despite some initial preservation immediately after IRE [52]. Current IRE parameters may temporarily preserve endocrine function [20], but do not prevent long-term loss of beta cell integrity. The present results indicate there is a therapeutic window immediately after ablation, during which adjuvant regenerative therapy or anti-inflammatory agents may be administered to support and sustain beta cell function.

The proliferative response observed after 2 days of treatment, as indicated by increased Ki-67 staining, suggests activation of intrinsic pancreatic regeneration pathways [53]. However, the observed regenerative capacity appears to be insufficient to completely compensate for the evident fibrosis and severe structural damage in the severely ablated region. Strategies to boost this regenerative potential by administering regenerative stimulants or anti-inflammatory agents should be a primary focus of future studies to enhance the clinical efficacy of IRE.

This study has several limitations that require further review. One of them was the unintended thermal impact. This suggests that careful interpretation of the results and further optimization of IRE parameters, including improved electrode cooling or temperature control measures, are needed to precisely distinguish between electroporation-induced and thermal-induced tissue effects. Another limitation is the limited sample size. The limited sample size restricted the ability to perform robust quantitative statistical analyses, which restricts the generalizability of our findings. Future investigations with larger cohorts will be required to validate and extend the current observations. In addition, the relatively short follow-up period of 2 weeks limited the comprehensive assessment of long-term fibrotic outcomes and regenerative potential. This short observation period also precluded evaluation of durable endocrine recovery. Importantly, our study did not establish a direct functional correlation between β-cell loss, fibrosis, and endocrine impairment, since functional assays such as glucose tolerance testing and serum insulin measurements were not performed. We have explicitly acknowledged this as a critical limitation and emphasized that future investigations should incorporate extended follow-up and functional endpoints to directly link histological alterations with sustained pancreatic endocrine function. Finally, although the porcine model closely resembles human pancreatic physiology, differences in immune responses and regenerative capacity require careful interpretation of clinical translation. The inclusion of other animal models and human clinical trials is crucial to generalizing the results of this study.

Furthermore, this study primarily relied on qualitative histological observations without extensive molecular and functional analysis, including blood glucose and serum insulin levels, detailed molecular quantification of apoptosis, fibrosis, and beta-cell preservation. The absence of such systemic and quantitative analyses limits the ability to directly link histological changes with pancreatic endocrine function. Future investigations should therefore integrate biochemical assays, quantitative image analysis, and standardized fibrosis scoring methods to provide a more robust and mechanistic understanding of IRE-induced tissue remodeling.

In summary, our findings were preliminary and exploratory, reflecting the limited sample size, short follow-up, and qualitative nature of the analysis. The results indicated selective pancreatic ablation by IRE while also revealing important challenges, including localized thermal injury, fibrosis progression, and incomplete regeneration. Large studies incorporating quantitative and functional endpoints were required to validate and expand these initial observations before clinical translation.

## 5. Conclusions

This study demonstrated that IRE enables selective pancreatic ablation while preserving adjacent tissue, but also induces fibrosis, β-cell loss, and localized thermal effects. These results highlight both the promise and the challenges of IRE in pancreatic tissue.

Given the small sample size, short follow-up, and qualitative design, the findings remain preliminary. Larger, long-term studies with functional endpoints are required to confirm these observations and guide optimization of IRE protocols for clinical translation.

## Figures and Tables

**Figure 1 biomedicines-13-02222-f001:**
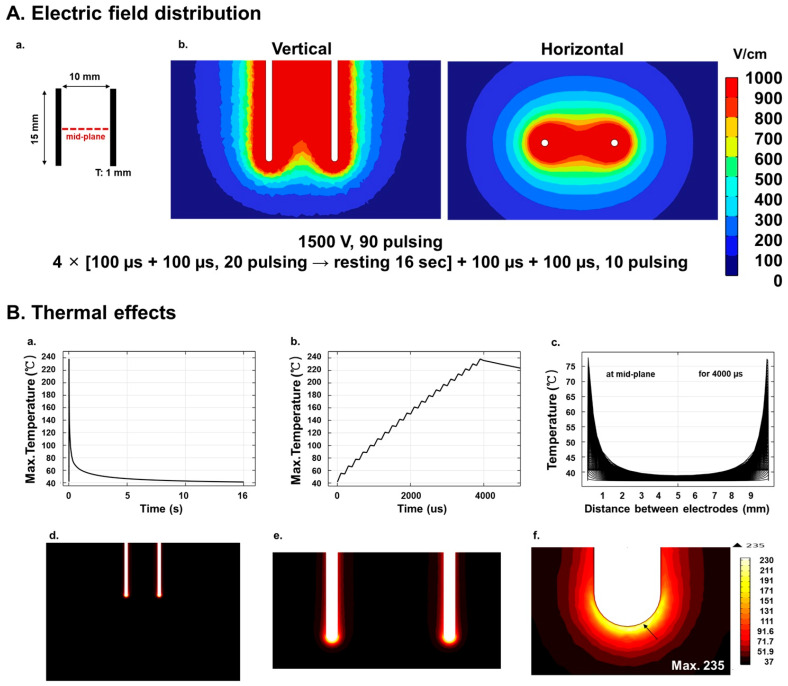
Simulated electrical field and thermal distributions during IRE treatment in porcine pancreatic tissue: (**A**) Electric field distribution. (**a**) Schematic diagram of the electrode configuration, showing two electrodes, each 15 mm in length, spaced 10 mm apart. The dashed red line indicates the mid-plane along which the temperature profile was extracted. (**b**) Simulated electric field distribution during IRE with an electric field strength of 1000 V/cm between electrodes. The IRE protocol comprised 90 pulses of 1500 V amplitude, 100 µs pulse width, and 100 µs inter-pulse interval, delivered in 4 sets of 20 pulses and one additional set of 10 pulses. A 16-second inter-set interval was applied to prevent sparking. (**B**) Thermal effects of electroporation. (**a**) Simulated maximum temperature profile over time after one set of pulses, showing a rapid rise to 235 °C at the electrode tip, followed by a cooling to 37 °C after the pulse. (**b**) Simulated temperature changes during a 4000 µs pulse cycle, highlighting the rapid temperature fluctuations at the electrode surface. (**c**) Simulated temperature gradient along the mid-plane between electrodes, extracted at 7.5 mm from the electrode tip during the 4000 µs pulse cycle, showing prominent thermal variations within the tissue, reaching a maximum of 78.4 °C. (**d**–**f**) Visualization of the temperature distribution showing the maximum temperature of 235 °C at the electrode tip (arrow indicates the location of the maximum temperature).

**Figure 2 biomedicines-13-02222-f002:**
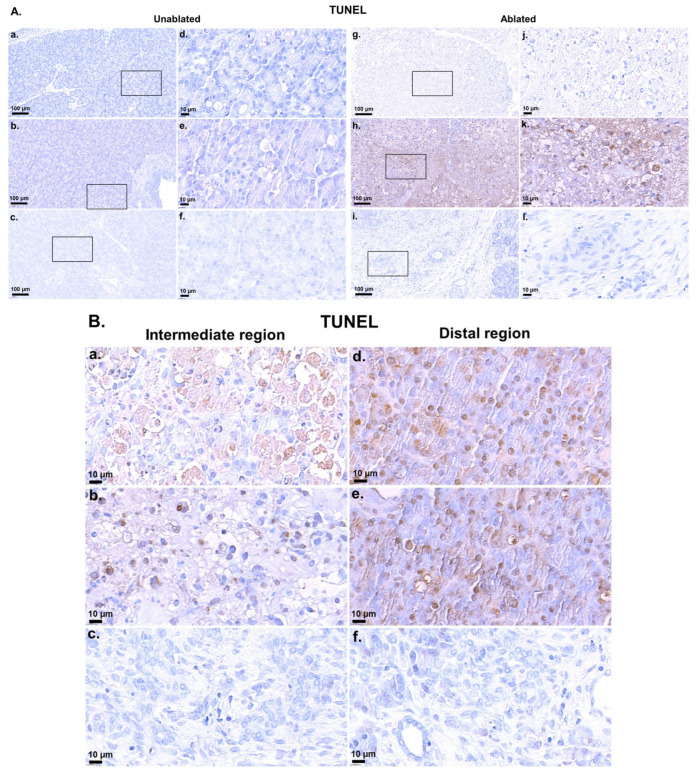
TUNEL assay for the pancreatic tissue after IRE treatment: (**A**) Representative images comparing the distribution of apoptosis in unablated (**a**–**c**) and ablated (**g**–**l**) pancreatic tissues at different time points after IRE treatment. Unablated tissues at 2 h (**a**,**d**), 2 days (**b**,**e**), and 2 weeks (**c**,**f**) post-IRE showed minimal apoptotic activity, whereas ablated tissues at 2 h (**g**,**j**), 2 days (**h**,**k**), and 2 weeks (**i**,**l**) post-IRE demonstrated apoptosis ranging from localized to extensive within the ablation zone. Scale bars: 100 µm (**a**–**c**,**g**–**i**) and 10 µm (**d**–**f**,**j**–**l**). The black boxes in the left figures indicate the enlarged regions in the corresponding right figures. (**B**) Detailed apoptosis in the intermediate and distal regions according to the electrode position. Apoptosis was observed at 2 h (**a**,**d**) and 2 days (**b**,**e**) after IRE treatment, but was negative at 2 weeks (**c**,**f**). Scale bars: 10 µm.

**Figure 3 biomedicines-13-02222-f003:**
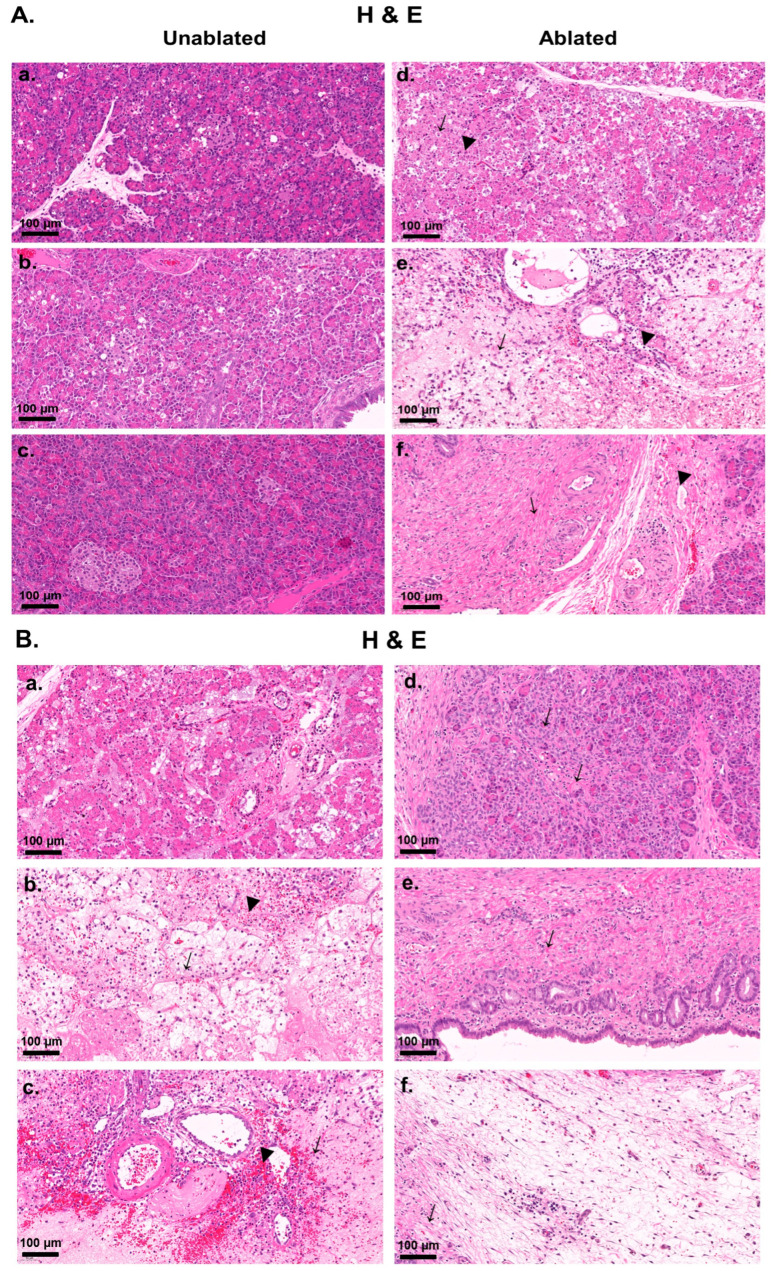
H&E staining of pancreatic tissues after IRE treatment: (**A**) Representative images comparing unablated (**a**–**c**) and ablated (**d**–**f**) pancreatic tissues at different time points after IRE. In ablated tissues: (**d**) apoptotic (↓) and necrotic (▼) cells after 2 h; (**e**) prominent necrosis (↓) and inflammation (▼) after 2 days; (**f**) heavy fibrosis (↓) and loss of pancreatic structure (▼) at 2 weeks after IRE. (**B**) Detailed histopathological changes after IRE in ablated tissues: structural disorganization after 2 h (**a**), necrosis (↓) and hemorrhage (▼) after 2 days (**b**,**c**), and structural remodeling and fibrosis (↓) after 2 weeks (**d**–**f**). Scale bars: 100 µm.

**Figure 4 biomedicines-13-02222-f004:**
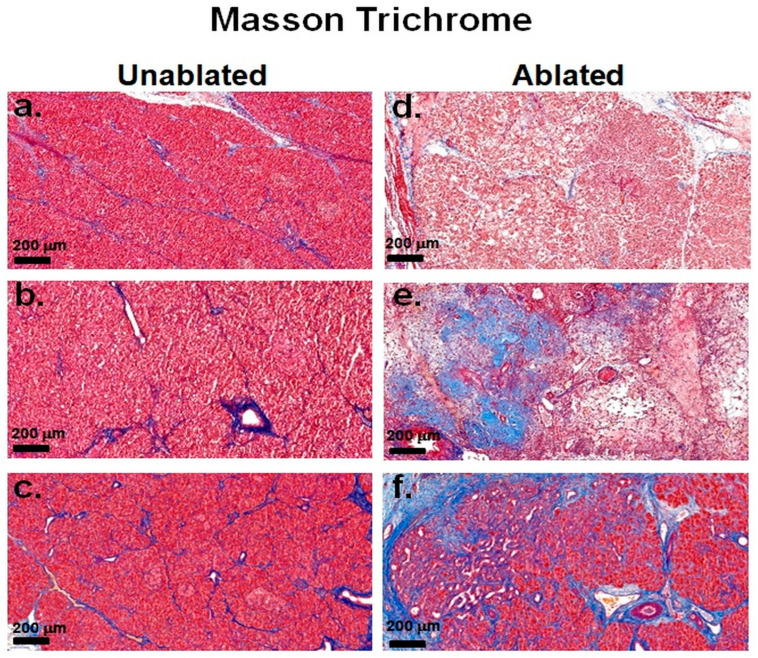
Masson’s trichrome staining of pancreatic tissue after IRE. Fibrosis was minimal, tissue architecture was preserved, and collagen deposition was very low at 2 h (**a**), 2 days (**b**), and 2 weeks (**c**) in the unablated regions. Fibrosis was minimal at 2 h (**d**), collagen deposition increased at 2 days (**e**), and fibrosis was prominent around the bile ducts and blood vessels at 2 weeks (**f**) in the ablated regions. Scale bars: 200 µm.

**Figure 5 biomedicines-13-02222-f005:**
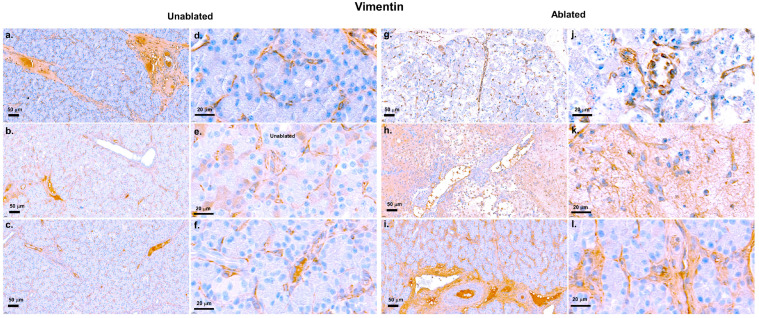
Vimentin staining of pancreatic tissue after IRE. Low vimentin expression limited fibroblast activity in the non-resected area at 2 h (**a**,**d**), 2 days (**b**,**e**), and 2 weeks (**c**,**f**). Low vimentin expression at 2 h (**g**,**j**), moderately increased vimentin staining at 2 days (**h**,**k**), and widespread vimentin expression at 2 weeks (**i**,**l**) in the ablated regions. Scale bars: 50 µm (**a**–**c**,**g**–**i**) and 20 µm (**d**–**f**,**j**–**l**).

**Figure 6 biomedicines-13-02222-f006:**
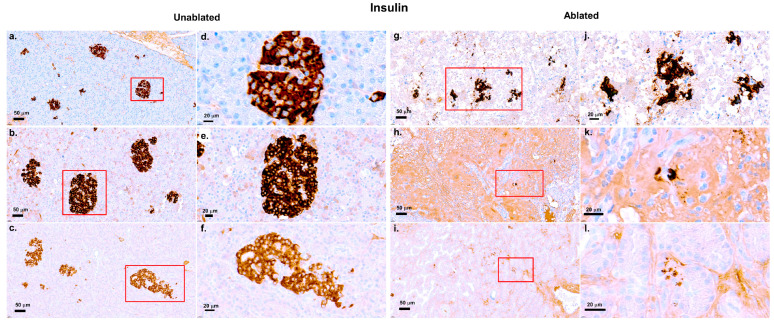
Insulin staining of pancreatic tissue after IRE. Well-preserved insulin-positive beta cells at 2 h (**a**,**d**), 2 days (**b**,**e**), and 2 weeks (**c**,**f**), with dense, intact cells clustered in the unablated regions. Dispersed insulin-positive cells and fragmented islet structures at 2 h (**g**,**j**), prominent reduction and further fragmentation at 2 days (**h**,**k**), and prominent reduced insulin-positive cells by 2 weeks (**i**,**l**) in the ablated regions. Scale bars: 50 µm (**a**–**c**,**g**–**i**) and 20 µm (**d**–**f**,**j**–**l**). The red boxes in the left figures indicate the enlarged regions in the corresponding right figures.

**Figure 7 biomedicines-13-02222-f007:**
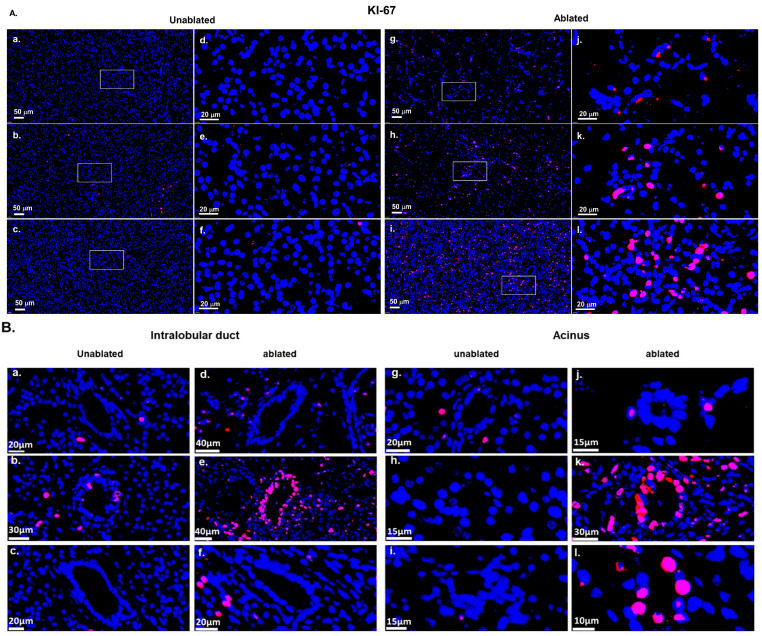
Ki-67 staining of pancreatic tissue after IRE: (**A**) Representative images of Ki-67 staining. Very low-level proliferation in the unresected area at 2 h (**a**,**d**), 2 days (**b**,**e**), and 2 weeks (**c**,**f**). Initial proliferation at 2 h (**g**,**j**), a prominent increase in Ki-67-positive cells at 2 days (**h**,**k**), and sustained elevated proliferation at 2 weeks (**i**,**l**) in the ablated regions. Scale bars: 50 µm (**a**–**c**,**g**–**i**) and 20 µm (**d**–**f**,**j**–**l**). White boxes in the left figures indicate the enlarged regions in the corresponding right figures. (**B**) Detailed proliferation in intralobular ductal and acinar regions. Minimal proliferation across all time points in the unablated tissues (**a**–**c**,**g**–**i**). Initial proliferation at 2 h (**d**), a peak at 2 days (**e**), and reduced but persistent activity at 2 weeks (**f**) in the ablated intralobular ducts. Initial proliferation at 2 h (**j**), a prominent increase at 2 days (**k**), and sustained elevation at 2 weeks (**l**) in the ablated acinar regions. Images were shown at varying magnifications to optimally visualize ductal acinar structure; therefore, different scale bars were used and indicated in each panel. Scale bars: 10–40 µm.

## Data Availability

The original contributions presented in this study are included in the article. Further inquiries can be directed to the corresponding authors.

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
