# Peer review of "Pancreatic Tissue Remodeling and Fibrosis After Irreversible Electroporation: A Histopathological and Thermal Perspective"

_biomedicines, 2025, doi:10.3390/biomedicines13092222_

Round 1

Reviewer 1 Report

Comments and Suggestions for Authors

This study addresses an important question regarding the tissue remodeling and fibrotic response following irreversible electroporation (IRE) in a porcine pancreas model. However, several methodological limitations, interpretative gaps, and presentation issues should be addressed before the manuscript can be considered for publication.

The abstract is somewhat dense; simplifying the description of methods would improve readability.   Several grammatical issues should be corrected.

The introduction repeats much of the background already covered in previous IRE reviews. This could be shortened to focus more on why this study is different.

Only seven pigs were included, with small groups (n=1 for early timepoints, n=3 for two weeks). This limited cohort reduces statistical power and makes it difficult to generalize the findings. The authors acknowledge this limitation, but the conclusions are presented too strongly given the sample size.

The simulation predicted extreme temperatures, that is to 235 °C at the electrode tip, yet in vivo measurement reported a peak of ~78 °C. This discrepancy raises questions about the accuracy of the thermal modeling parameters. The authors should clarify.

The fibrosis, apoptosis, Ki-67, and insulin changes are all described qualitatively. May consider including semi-quantitative data.

Loss of insulin-positive cells is interesting, functional readouts (e.g., glucose levels, serum insulin) would greatly help the claim.

Reviewer 2 Report

Comments and Suggestions for Authors

    This manuscript presents a well-designed preclinical study investigating the effects of irreversible electroporation (IRE) on porcine pancreatic tissue. The study addresses clinically relevant questions regarding tissue ablation, fibrosis, and regeneration, with potential implications for pancreatic cancer treatment. The methodology is robust, and the results are presented clearly. However, several areas require clarification, expansion, or refinement to enhance the manuscript's impact and scientific rigor.

  1. The introduction effectively contextualizes the clinical challenges of pancreatic cancer and the limitations of traditional thermal ablation techniques. However, the rationale for focusing on IRE's non-thermal effects could be strengthened by briefly mentioning prior conflicting evidence about thermal contributions in IRE.
  2. The thermal simulations (235°C at electrode tips) are striking and clinically significant. However, the manuscript does not fully reconcile these findings with the "non-thermal" premise of IRE.
  3. The "therapeutic window" for adjuvant therapies (e.g., antifibrotics) is a key insight but lacks specific mechanistic proposals.
  4. The short follow-up (2 weeks) and lack of long-term functional outcomes (e.g., glucose tolerance tests for endocrine function) are appropriately noted but could be expanded.
  5. Figures are well-annotated but could benefit from supplemental quantitative data (e.g., apoptosis/fibrosis indices).

Reviewer 3 Report

Comments and Suggestions for Authors

This manuscript addresses an important and timely topic: the effects of irreversible electroporation (IRE) on pancreatic tissue remodeling, fibrosis, and regeneration. The study combines in vivo porcine experiments with numerical simulations to explore histopathological and thermal changes following IRE. The results suggest that while IRE induces selective cell death and preserves adjacent structures, it also triggers significant fibrosis and thermal injury, which may limit functional recovery.

The study is generally well-written, logically structured, and scientifically valuable. However, there are several major and minor concerns that should be addressed before acceptance.

Introduction:

  1. The rationale for focusing on fibrosis and beta-cell preservation be strengthened by citing more recent clinical data. Although, mentioned, but weakly supported by outdated or preclinical citations.
  2. Porcine model selection should be justified in the introduction section,  present but buried in Methods section and not well-integrated into Introduction.
  3. Only 5 animals were analyzed, and the follow-up period was limited to 14 days. These constraints weaken the strength of conclusions about regeneration and fibrosis. Emphasize these limitations more strongly in the Discussion, and Conclusion like Abstract.

Methodology section:

  1. The manuscript uses two different section thicknesses (3 µm vs. 4 µm). In H&E staining (Section 2.5): “sectioned into 4 µm-thick slices” and In Ki-67 immunofluorescence staining (Section 2.9): “Tissues were sectioned to 3 µm thickness” needs proper justification why the section size is different for these two methods. Either be consistent with the size or put a one line justification. 
  2.  In the methodology section Figure 1, the study reports temperatures as high as 235 °C at the electrode tip and 78 °C in tissue, which contradicts the central claim of IRE as a non-thermal modality. This issue requires careful clarification: are these simulation artifacts, or do they reflect real clinical risks?
  3. The methodology discusses the Simulated electrical field and thermal distributions during IRE treatment but lack a dedicated portion to discuss how electrode were design.

Results:

  1. Many results including fibrosis, apoptosis and regeneration are presented qualitatively. Although the results are acceptable but a quantitative analyses (e.g., fibrosis scoring, apoptotic index, insulin-positive cell counts) can further  strengthen the study (Optional).

  2. The Figures are mostly descriptive and lacks statistical representation.
  3. In Figure 7B, the images (a–l) are presented with varying scale bars (10 µm, 15 µm, 20 µm, 30 µm, and 40 µm). The authors are advised to ensure consistency by using a uniform scale bar across all images.
  4. In addition, figure 7 should be placed before discussion.

Discussion:

  1. The “window for adjuvant therapy” concept after apoptosis should be highlighted more clearly in the discussion section.

Conclusion:

  1. A dedicated subheading for conclusion is missing.   

Round 2

Reviewer 1 Report

Comments and Suggestions for Authors

The study remains descriptive, without quantitative analysis or systemic functional assays. However, these are now clearly acknowledged as limitations rather than oversights. Some figures remain largely qualitative, but this is acceptable for an exploratory large-animal study.

The revision adequately addresses prior concerns, improves clarity, and appropriately tempers conclusions. Given that the paper is framed as an exploratory descriptive study, I find the revisions satisfactory and supportive of acceptance.

Author Response

We sincerely thank the reviewer for the careful evaluation and supportive comments. We appreciate the acknowledgement that the descriptive and qualitative aspects are acceptable within the context of an exploratory large-animal study. We are also grateful for recognizing our efforts to clarify the limitations and to refine the conclusions. We have no further modifications to add in response, but we highly value your positive assessment and constructive feedback.

Reviewer 2 Report

Comments and Suggestions for Authors

     This study investigates tissue responses in a porcine pancreas model following IRE, focusing on apoptosis, thermal effects, fibrosis progression, and beta-cell preservation. This study combines numerical modeling with histological analysis to explore both electroporation-induced and thermally mediated effects. The topic is relevant, as IRE is an emerging non-thermal ablation technique for pancreatic cancer, but questions remain regarding its fibrotic consequences and thermal contributions.

  1. The manuscript is generally well-structured, and the experimental approach is sound. However, the very small sample size (n=7 pigs, with one excluded) significantly limits the statistical power and generalizability of the findings.
  2. The simulations predict extreme temperatures (235°C at the electrode tip, 78.4°C in surrounding tissue). While the authors correctly note this may be an overestimation due to model constraints, these values are exceptionally high and somewhat difficult to reconcile with the established notion of IRE as a non-thermal modality. Some experimental studies report much smaller temperature increases.
  3. The study demonstrates progressive loss of insulin-positive beta cells and fibrosis but does not correlate these histological findings with pancreatic endocrine function. 
  4. A 14-day period is sufficient to observe acute apoptosis and early fibrosis but is too short to assess long-term fibrotic resolution or chronic functional impairment.
  5. The discussion could be slightly enhanced by briefly elaborating on how the observed fibrotic response might impact the treatment window for adjuvant therapies (e.g., immunotherapy1 or stem cell therapy28) following IRE ablation in pancreatic cancer.

Reviewer 3 Report

Comments and Suggestions for Authors

The revised manuscript, Pancreatic tissue remodeling and fibrosis after irreversible electroporation: a histopathological and thermal perspective, has been thoroughly evaluated. A comparative analysis with the previous version indicates significant improvements. The current revision incorporate/addresses all the concerns raised earlier and justification provided for some question are rational. The manuscript is now mature enough and suitable for publication in its present form.

Author Response

We sincerely thank the reviewer for the careful evaluation and positive feedback. We are grateful that you found our revisions satisfactory and that the manuscript is now considered suitable for publication. We truly appreciate your constructive comments throughout the review process, which have helped us to significantly improve the clarity and overall quality of our work.